# MicroRNA Profiling in Papillary Thyroid Cancer

**DOI:** 10.3390/ijms25179362

**Published:** 2024-08-29

**Authors:** Richard Armos, Bence Bojtor, Marton Papp, Ildiko Illyes, Balazs Lengyel, Andras Kiss, Balazs Szili, Balint Tobias, Bernadett Balla, Henriett Piko, Anett Illes, Zsuzsanna Putz, Andras Kiss, Erika Toth, Istvan Takacs, Janos P. Kosa, Peter Lakatos

**Affiliations:** 1Department of Medicine and Oncology, Faculty of Medicine, Semmelweis University, 1083 Budapest, Hungary; armos.richard@semmelweis.hu (R.A.); bojtorbence01@gmail.com (B.B.); lengyel.balazs.levente@semmelweis.hu (B.L.); kiss.andras1@semmelweis.hu (A.K.); szili.balazs@semmelweis.hu (B.S.); tobias.balint@gmail.com (B.T.); piko.henriett@semmelweis.hu (H.P.); barkaszine.dr.illes.anett@semmelweis.hu (A.I.); putz.zsuzsanna@med.semmelweis-univ.hu (Z.P.); takacs.istvan@semmelweis.hu (I.T.); kosa.janos@semmelweis.hu (J.P.K.); 2SE HUN-REN-TKI ENDOMOLPAT Research Group, 1085 Budapest, Hungary; balladetti@gmail.com; 3Centre for Bioinformatics, University of Veterinary Medicine, 1078 Budapest, Hungary; pappmarci95@gmail.com; 4Department of Pathology, Forensic and Insurance Medicine, Faculty of Medicine, Semmelweis University, 1091 Budapest, Hungary; illyes.ildiko@semmelweis.hu (I.I.); kiss.andras@semmelweis.hu (A.K.); 5Department of Surgical and Molecular Pathology, National Institute of Oncology, 1122 Budapest, Hungary; dr.toth.erika@oncol.hu

**Keywords:** miRNA, papillary thyroid carcinoma, thyroid cancer, miRNA pattern, sequencing, molecular diagnostics

## Abstract

Genetic alterations are well known to be related to the pathogenesis and prognosis of papillary thyroid carcinoma (PTC). Some miRNA expression dysregulations have previously been described in the context of cancer development including thyroid carcinoma. In our study, we performed original molecular diagnostics on tissue samples related to our own patients. We aimed to identify all dysregulated miRNAs in potential association with PTC development via sequencing much higher numbers of control-matched PTC tissue samples and analyzing a wider variety of miRNA types than previous studies. We analyzed the expression levels of 2656 different human miRNAs in the context of 236 thyroid tissue samples (118 tumor and control pairs) related to anonymized PTC cases. Also, KEGG pathway enrichment analysis and GO framework analysis were used to establish the links between miRNA dysregulation and certain biological processes, pathways of signaling, molecular functions, and cellular components. A total of 30 significant differential miRNA expressions with at least ±1 log_2_ fold change were found related to PTC including, e.g., miR-551b, miR-146b, miR-221, miR-222, and miR-375, among others, being highly upregulated, as well as miR-873 and miR-204 being downregulated. In addition, we identified miRNA patterns in vast databases (KEGG and GO) closely similar to that of PTC including, e.g., miRNA patterns of prostate cancer, HTLV infection, HIF-1 signaling, cellular responses to growth factor stimulus and organic substance, and negative regulation of gene expression. We also found 352 potential associations between certain miRNA expressions and states of clinicopathological variables. Our findings—supported by the largest case number of original matched-control PTC–miRNA relation research—suggest a distinct miRNA expression profile in PTC that could contribute to a deeper understanding of the underlying molecular mechanisms promoting the pathogenesis of the disease. Moreover, significant miRNA expression deviations and their signaling pathways in PTC presented in our study may serve as potential biomarkers for PTC diagnosis and prognosis or even therapeutic targets in the future.

## 1. Introduction

MicroRNAs (miRNAs) form a special group of RNAs, being small and non-coding nucleic acids with a fine-tuning regulatory role in gene expression [1]. Briefly, miRNAs are around 18–25 nucleotides in length and take effect by binding to their complementary sequences on messenger RNA (mRNA) transcripts, which could result either in target degradation or translational repression and, by that, gene silencing [2]. miRNAs are considered to be quite stable molecules, making them accessible from a variety of biological samples such as fresh tissues, fine-needle aspiration biopsy (FNAB) specimens [3,4], blood samples [5,6], or even from formalin-fixed paraffin-embedded (FFPE) tissues [7]. miRNAs represent a long-hidden and complex layer of gene regulation that is a crucial component in many physiological and pathological processes including cardiovascular diseases (e.g., cardiac fibrosis and cardiac hypertrophy) [8,9], microbiome homeostasis [10], diabetes mellitus [11,12], calcium and bone metabolism [13,14], as well as in the pharmacodynamics of certain drugs [15]. miRNAs influence many cellular processes such as differentiation, proliferation, apoptosis, and metastasis development [16,17,18,19]. In addition, research on the role of miRNAs in anti-cancer drug sensitivity in thyroid cancer has also been published [6].

In oncology, miRNAs are receiving more and more attention due to their duality acting both as oncogenes and tumor suppressors [20]. Emerging data suggest that different molecular backgrounds, including aberrant miRNA expressions, are a common feature of various cancers [21,22], including thyroid malignancies and their most common form, PTC, in particular [23]. PTC is seemingly well known by many clinicians and pathologists, although it is still slowly turning into a more heterogeneous category, which can be further divided not only into clinicopathological subtypes but molecular ones as well. Admittedly, some papers have already been published investigating the role of miRNAs in PTC pathogenesis [24,25,26]; however, establishing the exact depth of the correlation between specific miRNA expressions and the development of the disease lacks original molecular studies with sufficiently large control case numbers even to this day. Also, these studies provide markedly smaller coverage of miRNA types. For instance, even The Cancer Genome Atlas (TCGA) project involved PTCs with only 59 matched controls for miRNA analysis and evaluated 1046 different miRNAs while lacking clinicopathological aspects [24]. miRNAs’ association with the pathogenesis of other conventional thyroid cancer types such as follicular thyroid carcinoma (FTC), medullary thyroid carcinoma (MTC), or anaplastic thyroid carcinoma (ATC) has also been noticed [27]. In cases when differentiated thyroid carcinoma (DTC) (including PTC and FTC) was diagnosed, tumor development might be traced back to a DICER1 RNase IIIb hotspot mutation generating an unbalanced expression ratio of 5p:3p miRNAs [28].

PTC, characterized by its unique molecular signature, often exhibits alterations in miRNA expression [19,21,29]. The link between these miRNA deviations and disease development is making specific miRNAs perfect candidates for being diagnostic or prognostic markers. Indeed, miR-21, miR-127, miR-136, miR-146b, miR-221, and miR-222 are frequently upregulated in PTC and are associated with a more aggressive course of the disease and an overall poorer prognosis [6,19,22,24,27,30,31,32,33,34,35,36,37]. Also, *BRAF*-targeting miR-486-5p, miR-9-5p, and miR-708-3p in PTC were previously described as downregulated and—in case of miR-486-5p—associated with lymph node metastasis development as well as more advanced tumor stage and recurrence risk [25]. Based on previous findings, the 5p/3p expression ratio of specific miRNAs might also play a role in the onset of lymph node metastases [26]. Upregulation of miR-181a in thyroid cancer has also been identified previously [19]. Moreover, expression alterations of miRNAs such as miR-204, miR-146b, miR-221, and miR-222 have been described in association with important cancer features like extrathyroidal invasion and/or metastasis development [17,19,36,37]. Alternatively, a decrease in tumor-suppressing miRNA levels, such as let-7, miR-125b, and miR-204-5p [24,38], has also been noted, propagating the malignant conversion of thyroid cells [39]. Furthermore, miR-136, miR-21, and miR-127 showed significant correlation with the risk of persistent disease and potential relapse as well as with the risk categories defined by the American Thyroid Association’s (ATA) risk stratification system [40,41].

This study aims to reveal differences in miRNA expression patterns between cancerous and non-cancerous thyroid tissue samples related to patients previously diagnosed with PTC. In respect of previous findings, a better understanding of miRNA expression profiles of malignancies—including thyroid cancer—may provide insights into tumorigenesis and disease progression as well as potential therapeutic targets in the future [35,37]. For this purpose, we conducted a comprehensive molecular investigation to detect specific miRNAs associated with PTC based on their expression differences between tumor sections and adjacent non-tumor tissues. We evaluated the miRNA profiles of 236 individual thyroid tissue samples (one cancerous and one healthy sample per case) related to 118 patients previously diagnosed with PTC and analyzed the sequencing data of the samples in the context of 2656 types of human miRNAs, doubling the control-matched sample size and more than doubling the examined miRNA cluster size compared to the TCGA study [24].

## 2. Results

### 2.1. Study Population

miRNAs were isolated from 258 tissue samples (related to the cancerous and normal samples of 129 PTC patients). Ten tissue sample pairs were excluded due to inapplicable isolate specimens; the remaining ones were then forwarded to sequencing. Via sequencing, expression patterns were examined in the context of 2656 different types of miRNAs in total. During the bioinformatic assessment following sequencing, one more sample pair was also excluded due to insufficient sequencing yield. After all necessary exclusions, tumor and control sample pairs related to a total of 118 PTC patients were included in the evaluation (Figure 1). The tumor samples included in our study contained the following histological subtypes of PTC: conventional (n = 96), follicular subtype (n = 16), oncocytic (n = 4), columnar cell (n = 1), and Warthin-like (n = 1) (Table 1).

### 2.2. Descriptive Analysis

The descriptive analysis of miRNA expression profiles in PTC patients provided a detailed landscape of miRNA dysregulation of the disease. We listed miRNAs with significant expression dysregulation in PTC in Table 2 based on our sequencing data. Subsequent analysis following isolation revealed significant differences in the expression of 30 individual miRNAs in total, 27 of them being over-represented (e.g., miR-551b-3p, miR-146b-3p, miR-9983-3p, miR-221-3p, miR-222-3p, and miR-375-3p) and 3 of them being under-represented (miR-206, miR-873-3p, and miR-204-3p) in PTC compared to the healthy parts of the thyroid. As mentioned above, we presented the results indicating not only statistical significance but individual fold changes as well. Those “top 20 miRNAs” with the highest significance are depicted in Figure 2. miR-551b-3p, miR-146b-3p, miR-146b-5p, miR-221-3p, miR-375-3p, miR-873-3p, and miR-204-3p are the most noteworthy. It is also worth mentioning that 582 of the studied miRNA types were not expressed in thyroid tissue at all.

Our volcano plot (Figure 3) illustrates even more effectively the differential expression of miRNAs, with the extent of expression deviation (log_2_ fold change) on the horizontal axis and the strength of significance (−log_10_P) on the vertical axis. miRNAs of special interest (both statistically significant and with fold change above the threshold as represented in Table 2) are marked with red dots and located within upper left (representing underexpression) and upper right (representing overexpression) parts of the figure.

Differential expression in relation to the 30 “top miRNAs” clustered together based on their strong significance can also be clearly seen in Figure 4C,D. It is noticeable that cancerous tissue samples (red group) (A) tend to heavily overexpress or underexpress the “top miRNAs” compared to healthy controls (blue group) (A). Expression deviations of each “top miRNA” for all samples are displayed by the color codes of the Z-score (B), with red cells of the heatmap representing miRNA overexpressions and blue ones underexpressions.

We also conducted a miRNA expression comparison between the conventional and follicular subtypes within our PTC tumor sample cohort. We were unable to detect any significant miRNA expression differences between these two subtypes in the context of the “top miRNAs”. We did not perform further analysis of other subtypes due to their low frequency in the study population.

### 2.3. Principal Component Analysis of Every Studied miRNA and Those with Significant Expression Differences

Via PCA of miRNA expression profiles, we detected clearly distinct patterns that differentiate between control and tumor samples. PCA plot (A) (Figure 5) encompasses all evaluated miRNAs and reveals that the first two principal components (PC1 and PC2) account for a significant portion of the variance (44.27% and 17.78%, respectively) within the dataset. The scatter of control samples (red) and tumor samples (blue) along the axes shows a discernible but overlapping distribution, indicating only a nuanced relationship between miRNA expression and tumor status. However, by enhancing the analysis via focusing solely on miRNAs that exhibit significant expression differences, PCA yields a starker contrast between the control and tumor groups as shown in plot (B) (Figure 5). In this refined scenario, PC1 alone captures an impressive 86.07% of the variance. Here, the separation between the red and blue dots is much more pronounced, suggesting that the miRNAs labeled as significant ones could serve as robust biomarkers for PTC. PCA comes with the ability to visualize our massive data pool and presents the different miRNA expression landscapes between PTC and healthy thyroid tissue in a transparent way.

### 2.4. KEGG Pathway Analysis and Gene Ontology Biological Process, Cellular Component, and Molecular Function Enrichment Analyses Based on Statistically Significant (p ≤ 0.05) miRNAs (ORA—Over-Representation Analysis)

The KEGG pathway [42,43] and Gene Ontology [44,45,46] analyses revealed a set of molecular and biological processes heavily enriched in conjunction with the significant miRNAs. The results suggest an underlying complex network of miRNA-mediated regulations that extends beyond PTC pathogenesis. The enrichment of certain pathways, such as those related to cancer signaling, underscores the potential roles the investigated miRNAs may play in the development of PTC and probably thyroid cancer in general. It also highlights the functional consequences of miRNA dysregulation (Figure 6).

In addition, overlaps with pathways implicated in other diseases provide insights into a likely shared molecular origin of pathogenesis and could be the basis of future research related to the diagnostics of thyroid cancer comorbidities as well as the systemic effects of PTC-associated miRNA alterations.

### 2.5. Association Analysis between miRNA Expressions and States of Selected Clinicopathological Variables

In our extensive analysis of miRNA expression data, we identified a total of 352 significant (*p* < 0.05) miRNA expression differences between different states of the examined clinicopathological variables—such as age, sex, ATA risk score, as well as TNM and AJCC (eighth edition) stages—of the study cohort. We investigated the miRNA expression deviations and their connection to clinicopathological variables in the cases of both tumor and adjacent healthy control tissues.

Notably, 36 of the miRNA expression deviations pertained to tumor samples, while the majority, 316, were related to control samples. Our analysis also revealed a quite balanced distribution in the context of the direction of miRNA expressions, with 165 links being caused by miRNA overexpression and 187 by underexpression. Furthermore, 31 of these “miRNA expression–state of variable” associations were highly intense, exhibiting extreme log_2_FC values either above 10 or below −10, as labeled explicitly in Figure 7. Among these stronger associations, it is worth highlighting those miRNAs with the most prominent links to ATA risk: miR-6880-5p (direct, in tumor), miR-6753-5p (inverse, in tumor), miR-3648 (inverse, in control), and miR-6862-3p (inverse, in control); and those showing link to TNM score: miR-6753-5p (inverse, in tumor), miR-6805-5 (inverse, in control), miR-519c-3 (inverse, in control), and miR-6862-3p (inverse, in control). Note that downregulation of miR-6862-3p in healthy thyroid tissue adjacent to PTC is associated with greater ATA risk score, TNM score, and clinical stage as well.

## 3. Discussion

In this comprehensive study, we identified 30 miRNAs that showed significant up- or downregulation in PTC compared to healthy thyroid tissue, reinforcing several miRNAs as promising biomarkers and suggesting their role in PTC pathogenesis. PCA also demonstrated clear separation of cancer versus healthy tissue, supporting the power of miRNA patterns in oncology diagnostics. As far as we know, this is the first study describing the association of PTC with the overexpression of miR-9983-3p, miR-4695-3p, miR-1277-5p, miR147b-3p, miR-511-3p, and miR-137-3p. miR-551b-3p was overexpressed in our cancer samples, suggesting its oncogenic role in PTC, unlike in other cancer types discussed in previous research. In consensus with the literature, we also confirmed the overexpression of miR-221, miR-222, and miR-146b, among others. In addition, not yet published associations between the miRNA pattern of PTC and other physiological events, biological processes, and diseases were revealed as well.

Despite being a well-researched topic, the exact pathogenesis of PTC remains unknown, with most of the current, mutation-based models providing only a partial understanding of the disease [47]. Therefore, deeper insights into non-conventional tumor formation caused by miRNA regulatory mechanisms on gene expression are required for the improvement of cancer diagnostics and therapeutics [48,49]. The role of miRNAs in the development of PTC is rather complex, but association with many signaling molecules such as tumor protein p53 (TP53) [21], cyclin-dependent kinase inhibitor 1B (CDKN1B) [21] insulin-like growth factor binding protein 5 (IGFBP5) [38], transforming growth factor beta (TGF-β) [50], zinc and ring finger 3 (ZNRF3) [35], the RB1 gene [19], as well as serum thyroglobulin (Tg) [5] levels is assumed.

Although the initial discovery of the disease as well as its treatment options are already quite effective with low rates of recurrence and complications, further improvement in the follow-up of the patients could still be achieved [51]. In clinical practice, disease management is heavily dependent on risk assessments such as the ATA risk stratification system, the latest version of which already takes the *BRAFV600E* mutation into account to estimate the chance of disease recurrence [41]. However, it lacks the inclusion of other genetic alterations and miRNA expression deviations, both of which could largely alter the long-term outcome of each clinical setting. For example, oncogenic cellular pathways like those related to *RET* or *NTRK* gene mutations can already be targeted by selective tyrosine kinase inhibitors (TKIs); some miRNAs such as miR-146b, miR-203a, miR-204, miR-221, or miR-222 are also suggested to be potential prognostic indicators [5,34,52,53], although available data in this regard are still controversial [54]. The latest studies on miRNAs suggest that an underlying correlation with *BRAF* mutations is possible as well [31,35]. It is also important to be aware that the currently used circulating biomarkers in DTC diagnostics and surveillance like Tg levels have limitations, especially in the presence of Tg antibodies (TgAb), which can heavily compromise the accuracy of Tg measurements [55]. This is why novel circulating biomarkers for diagnostics and surveillance are keenly researched [5,6], with thyroid stimulating hormone receptor (TSHR) mRNA, Tg mRNA, and certain miRNAs as candidates [55]. Messenger RNAs are inherently unstable molecules; in addition, the sensitivity of circulating mRNAs is dependent on the timing of blood sampling, and the specificity of Tg mRNA can easily be influenced by non-thyroid origins of the molecule and/or technical difficulties of the measurement [55]. In this sense, the inclusion of specific miRNA expressions in risk assessment might be worth considering.

Given their potential, our research aimed to detect and analyze the expression levels of a wide range of miRNA types in PTC, both in tumor tissues as well as in their adjacent healthy-tissue counterparts from the same patient’s thyroid. Firstly, we established two cohorts of identical size (tumor and control) and performed the same molecular analysis on each of them. With this approach, we were able to identify PTC-specific miRNAs expressed in significantly higher or lower amounts than in the control samples. This is consistent with previous findings in which miRNAs acted as promising diagnostic biomarkers distinguishing thyroid cancer from benign thyroid disease or healthy controls [5,6,34,36,56]. By comparing the molecular dynamics within the same thyroid source tissue, we also had the opportunity to eliminate most of the biases related to patient selection and sample processing.

Interestingly, we found that 30 individual miRNAs showed significant deviations from values in the control healthy tissue. Among these 30 miRNAs, we are the first to describe miR-9983-3p, miR-4695-3p, miR-1277-5p, miR147b-3p, miR-511-3p, and miR-137-3p in relation to PTC, although most of them have been previously mentioned in the context of other malignant diseases [57,58]. In the case of miR-9983-3p and miR-147b-3p, however, there is very little historical evidence regarding their roles in any cancer. Further studies on these novel PTC-associated miRNAs may help expand our knowledge of PTC.

In addition, our results demonstrated PTC-related up- and downregulation of specific miRNAs, some of which have already been mentioned in previous studies, but some have attracted less attention so far. The most relevant one of such miRNAs is miR-551b-3p, which exhibited the most substantial fold change in PTC when compared to adjacent healthy thyroid tissue. Notably, an almost 60-fold overexpression of miR-551b-3p was seen within PTC tissue (Figure 2). Indeed, previous papers have reported on the irregular expression of miR-551b-3p in PTC [59,60]. It is noteworthy, however, that miR-551b-3p has been formerly recognized rather as a tumor suppressor in malignancies, such as gallbladder or gastric cancers [61,62]. It is also worth mentioning that in these studies [59,60,61,62], the number of patients involved was much lower (n = 42–60) than in our investigation. Consistent with its tumor-suppressive effects, miR-551b-3p was found to be underexpressed in these malignancies. However, our finding, indicating rather a significant overexpression in malignant thyroid tissue, suggests a potential oncogenic role for miR-551b-3p in relation with PTC. This observation underscores the dynamic nature of miRNA function, wherein certain miRNAs may manifest either oncogenic or tumor-suppressive properties depending on the specific malignancy under consideration.

In agreement with previous data [31,63], our results also demonstrate the significant over-representation related to miR-21, miR-146b, miR-221, and miR-222 within PTC. Therefore, the potential oncogenic role of these miRNAs can be underscored. Furthermore, miR-146b has been previously associated with epithelial–mesenchymal transformation and the rather invasive features of PTC [35]. Our data confirm miR-146b as a highly promising and novel diagnostic factor in the context of PTC. Additionally, our investigation provides further validation of the conspicuous overexpression of miR-221 and miR-222 as well. Dysregulation of these miRNAs, recognized as among the most established ones in thyroid cancer, continues to be implicated in the molecular landscape of PTC [32]. This supports their significance as noteworthy factors in the pathogenesis of PTC, contributing to a more comprehensive understanding of the molecular mechanisms underlying thyroid cancer.

Differential miRNA expression between PTC and healthy thyroid tissue became even more apparent after we carried out a principal component analysis (PCA), which demonstrated a clear separation between the two cohorts of interest. This not only reinforces the validity of the recognized “top miRNAs” as promising biomarkers but also suggests their utility in distinguishing between different stages or subtypes of PTC. However, an overlap in PCA plots can be observed, indicating a complex interplay of miRNAs, which may reflect the heterogeneity of the disease as well as the relevance of a pattern-based approach during analysis instead of the evaluation of individual miRNA quantities.

In order to put our results into a broader context, we conducted an extensive enrichment analysis of miRNA expression profiles in PTC using both the KEGG pathway and GO term annotations. Our findings reveal a significant correlation between the dysregulated miRNAs and various biological pathways and processes that may contribute to the development of PTC and/or the coincidence of comorbidities with similar molecular background. In this regard, among other diseases and gene functions, we were able to demonstrate remarkable similarities between the miRNA pattern of PTC and that of prostate cancer, HTLV-infection, HIF-1 signaling, negative regulation of gene expression, as well as cellular responses to growth factor stimulus and to organic substance. Furthermore, based on our data comparison with GO cellular component database, miRNA dysregulation in PTC seems to be mostly influenced by the molecular changes of the protein-containing complex and the cytosol, as well as enzyme binding and transcription factor binding mechanisms.

As part of the study, we analyzed miRNA expression data in the context of different states of some clinicopathological variables related to the patients. We identified 352 significant “miRNA expression–state of variable” links, of which 31 were highly suggestive of being caused by underlying correlations between certain miRNA expression patterns and the presence of different clinicopathological states such as those related to ATA risk (miR-6880-5p, miR-6753-5p, miR-3648, and miR-6862-3p), TNM score (miR-6753-5p, miR-6805-5, miR-519c-3, and miR-6862-3p), and clinical stage (miR-6862-3p). miR-6862-3 underexpression in the healthy adjacent thyroid tissue showed a higher state of all these three variables. This underscores the notion that some miRNAs might have a role in the development of PTC, its clinical behavior, and the prognosis of the disease through direct or indirect effects (e.g., expression dysregulation facilitated by age or sex).

Our results not only enhance our understanding of PTC’s molecular foundations but also illuminate the possibilities of miRNAs as novel therapeutic targets and biomarkers.

## 4. Materials and Methods

### 4.1. Study Population, Sample Collection, and Histopathological Processing

PTC tissue samples of 129 anonymized cases—of which 118 cases were included in the final evaluation—who had consecutively undergone thyroid surgery and had previously been diagnosed and/or treated in our institution (Department of Medicine and Oncology, Semmelweis University, Hungary) were selected using clinical and histological databases. Then, the related clinicopathological data of the anonymized cases and their tissue blocks were collected from medical records of the Department of Medicine and Oncology, Semmelweis University, Hungary, and histopathological archives located at the Department of Pathology, Forensic and Insurance Medicine, Semmelweis University, Hungary, respectively. In this study, we included the following clinicopathological data as variables: age, sex, ATA risk score, TNM stage, and cancer stage based on the 8th edition of the American Joint Committee on Cancer (AJCC). The collected tissue samples were FFPE blocks from which hematoxylin–eosin-stained probe sections were performed to confirm tumor presence and histological type and to determine the percentage of tumor volume. Only those PTC samples which had a sufficient proportion of tumor burden at the site of interest within the FFPE block relative to the overall tissue volume were involved in further processing. This assessment involved conducting a trial section on the FFPE block, which was then stained with hematoxylin and eosin and examined under a microscope by a pathologist to exclude samples containing less than 80% relative tumor volume. Also, the collection of the non-tumor control samples was optimized by targeting tissue areas as far from the PTC localizations as technically feasible, usually the contralateral lobe of the thyroid.

Further preparation of the carefully selected tumor tissue blocks was carried out using macrodissection producing 4 pieces of 10 µm thick sections per sample, which were subjected to molecular diagnostics. An area containing tumor as well as normal tissue was dissected from the same sample, thus resulting in 129 × 2 specimens.

### 4.2. Molecular Processing (miRNA Isolation, Quality Control (QC), miRNA Quantification, and Sequencing)

Zymo Quick RNA FFPE kit (Zymo Research, Irvine, CA, USA) was employed for the isolation of miRNAs from the prepared FFPE tissue sections. The process started with the removal of paraffin using proprietary deparaffinization solution, which was followed by rehydration of the tissue. The tissue was then subjected to proteinase K digestion at 55 °C for 2 h and subsequently at 65 °C for an additional 15 min to ensure thorough lysis. After digestion, the samples were treated with RNA lysis buffer, which facilitated the selective binding of miRNAs to the kit’s Zymo-Spin IICR column. The column was then washed multiple times to remove contaminants. To ensure the elimination of genomic DNA, the samples were treated with DNase. Finally, the miRNAs were eluted in 50 μL of elution buffer. During molecular analysis, we performed RNA quality assessment. Initially, we determined RNA concentrations using the Qubit™ HS RNA Assay Kit (Thermo Fisher Scientific, Waltham, MA, USA) on a Qubit™ 3.0 fluorometer (Thermo Fisher Scientific, Waltham, MA, USA). When necessary, the concentration of miRNAs was measured again using the Qubit™ microRNA Assay Kit (Thermo Fisher Scientific, Waltham, MA, USA) also on a Qubit™ 3.0 fluorometer.

Then, we prepared miRNA libraries in the following multi-step process using NEXTFLEX^®^ Small RNA-Seq Kit v4 (PerkinElmer Inc. Waltham, MA, USA). We started with an input of 50 ng of RNA, followed by the ligation of the NEXTFLEX^®^ 3′ adenylated adapters. After that, we removed the excess 3′ adapters, then we ligated the NEXTFLEX^®^ 5′ adapters. This was succeeded by the reverse transcription, first-strand synthesis; post-synthesis, we conducted bead cleanup, then polymerase chain reaction (PCR) amplification was performed using barcoded primers (19 cycles). Lastly, we finished miRNA library preparation with size selection and cleanup.

The next step was the quality control of the miRNA libraries involving DNA concentration measurement using the Quant-iT™ 1X HS dsDNA Assay Kit (Thermo Fisher Scientific, Waltham, MA, USA) on either a FLUOstar Omega fluorometer (BMG Labtech, Ortenberg, Germany) or a Qubit™ 3.0 fluorometer, along with assessment of the fragment sizes using the LabChip^®^ GX Touch™ nucleic acid analyzer (PerkinElmer Inc. Waltham, MA, USA) with an HT DNA X-Mark Chip (CLS144006) (PerkinElmer Inc. Waltham, MA, USA) with the HT DNA NGS 3K Reagent Kit (PerkinElmer Inc. Waltham, MA, USA).

For pooling the libraries, we calculated the molar concentrations based on the overall concentrations and the fragment sizes. Then, equal molar quantities were pooled from the libraries. The concentration of this pool was measured using the Quant-iT™ 1X HS dsDNA Assay Kit again and diluted to the final concentration of 2 nM.

Finally, the sequencing was carried out on a NextSeq 2000 system (Illumina Inc., San Diego, CA, USA) using a P3 (1 × 50 cycles) kit. This setup allowed us to generate 2 × 40 bp paired-end reads. miRNAs showing expression levels below the set threshold of the applied NGS platform both in the PTC and control samples were considered as not-expressed miRNAs. During the process, we maintained the seeding concentration at 650 pM to ensure optimal sequencing depth and quality.

### 4.3. Data Analysis via Bioinformatics and Statistical Evaluation

The quality check of the raw reads was performed via FastQC v0.11.7 and MultiQC. Forward and reverse reads were merged via PEAR v0.9.11 and then quality trimmed with Trim Galore v0.6.10. The quality threshold was set to 30, and only reads between 18 and 30 base pairs (bp) in length were used for further analysis based on the literature. One sample pair was removed from the analysis due to insufficient sequencing yield compared to the other samples. Based on recommendations, Bowtie1 v1.3.1 was used for the alignment of the reads to the miRBase v22.1 *H. sapiens* miRNA database with the following parameters: -n 0 -l 8 --best --strata -m 1 -no-unal. Read counts were calculated for each miRNA using SAMtools v1.14.

The expression levels of miRNAs can span several orders of magnitude, making the direct comparison of raw data challenging and less informative. To address this, we presented our data using logarithmic values, specifically log_2_ fold change for expression levels and −log_10_P for *p*-values. We applied a threshold for significant differential expression with a minimum log_2_ fold change of ±1 and a *p*-value below 0.05 for determining the statistical significance. By using a logarithmic transformation, we were able to mitigate the impact of extreme values, create a symmetrical view of up- and downregulation (both of which can be relevant), and make our results more visible. Principal component analysis (PCA) is another statistical tool we applied with which we were able to reduce data dimensionality by identifying the most important patterns that describe the data variability. PCA also reorients the data into principal components, which are new, uncorrelated variables ordered by their importance.

Statistical analysis of the read counts was performed in the R v4.2.1 programming environment. Differential miRNA expression was calculated with the DESeq2 package. miRNAs were evaluated to be significantly differentially expressed if the absolute value of the estimated log_2_ fold change was higher than or equal to 1 and the Benjamini–Hochberg-corrected *p*-value was less than or equal to 0.05. The ComplexHeatmap, EnhancedVolcano, and ggplot2 packages were used for the data visualization. The network graph was constructed using Python’s NetworkX library.

To highlight the potential biological and clinical implications of miRNA dysregulation in PTC, we performed a comprehensive KEGG pathway enrichment analysis using the miEAA analysis server, leveraging the miRPathDB database; in addition, we utilized the Gene Ontology (GO) framework to systematically categorize the functions of genes influenced by the differentially expressed miRNAs. The GO Resource is a collaborative bioinformatics tool that provides consistent descriptions of gene products across databases and species. It encompasses structured networks of defined terms that represent gene product properties, covering biological processes, cellular components, and molecular functions. miEAA v2.1 was used for an over-representation analysis on the miRNAs with adjusted *p*-values less than or equal to 0.05. The analysis was performed on the Gene Ontology and KEGG terms available in the miRPathDB database via the miEAA analysis server. Only terms with at least 10 genes were surveyed, and they were considered significant if the FDR-corrected *p*-value was less than 0.01.

### 4.4. Literature Review

Our study utilized the miRBase v22.1 *H. sapiens* miRNA database for human miRNA sequence and annotation retrieval. The literature search was conducted for miRNA-related disease associations and biological interactions through NCBI’s PubMed database, ensuring the inclusion of the most recent and relevant studies up to the date of access. We aimed to select peer-reviewed articles containing relevant data about the miRNAs of our interest, particularly in the context of human diseases. The literature search was conducted in the period between 20 November 2023 and 20 January 2024. Multiple synonymous search terms were concurrently utilized in order to find the most relevant data in the existing literature. These search terms included “papillary thyroid carcinoma”, “PTC”, “thyroid carcinoma”, “miRNA”, and “microRNA” as well as the names of individual miRNAs.

## 5. Conclusions

Our investigation on PTC–miRNA interactions involving the largest number of original molecular data resulting from PTC samples with matched controls (n = 118-118) and the most comprehensive set of analyzed miRNAs (n = 2656) provides strong evidence that the miRNA expression profiles manifest differently in tumorous and non-tumorous areas of the thyroid gland in the case of PTC by a significant margin. The data also suggest the essential role of the identified key miRNAs in PTC pathogenesis, their contribution to different clinicopathological states, and the fundamental similarities between the molecular pattern of other biological processes and that of PTC. It is noteworthy that links between the expression levels of some miRNAs and values related to disease advancement (ATA risk, TNM, and clinical stage) can be found not only when analyzing PTC tissue itself but for the histopathologically healthy adjacent tissue as well.

Most of the time, miRNAs labeled as significant were rather overexpressed in PTC cancer tissue; however, some of them showed a significantly reduced expression in PTC. Moreover, 582 miRNAs showed no expression in either tumor or control samples; however, these miRNAs could still turn out to be clinically significant in future studies of other types of thyroid cancer such as follicular, medullary, anaplastic, or even non-invasive follicular thyroid neoplasm with papillary-like nuclear features (NIFTP). Our findings hold promise for the development of novel diagnostic and therapeutic strategies for PTC patients targeting molecular pathways related to specific miRNAs. Our results not only emphasize the importance of miRNA expression patterns in thyroid cancer research but also highlight that these small non-coding nucleic acids may contribute to personalized medicine in the future. The evidence of an underlying interplay of certain miRNAs with the tumor environment and immune cells is emerging and, therefore, worth the consideration of further studies in these directions. As shown in the cases of miR-221 and miR-222, miRNAs have an effect on the thyroid stroma not only when cancer develops but also in other diseases with more benign behavior, such as multinodular goiter [36]. In previous studies, miR-222-3p expression in thyroid cancer was also associated with immune microenvironment regulation [64]. However, comprehensive, in-depth studies regarding the miRNA-immune axis in thyroid cancer are still needed. For instance, PD-L1, a predictive biomarker related to immune response and cancer immunotherapy, has recently been reported to be associated with PTC [65]. Indeed, later investigation of any causal interactions between miRNA patterns and PD-L1 expression in PTC seems to be a promising direction. Furthermore, it would also be worth investigating the potential of all the PTC-related miRNAs of this study as possible liquid biopsy biomarkers resulting from the serum as it is already successfully presented with a few miRNAs in PTC as well as with other markers in other malignancies [7,66,67]. This could help diagnostics in the form of a routine laboratory test and could also provide a joint molecular diagnostic methodology for parallel research on different cancer types.

It should be noted that our study has some limitations. First and foremost, though, our study was designed as original molecular research; however, we utilized tissue samples retrospectively from existing histological archives. Individual miRNA functions and their exact roles in molecular pathways were not investigated in this research; we only compared expression deviations of each miRNA between cancerous and healthy thyroid tissues. Also, we did not consider miRNA relations to mutational data such as *BRAFV600E*. This restriction was mainly due to the limited amount of tissue samples suitable for molecular processing available in our archives. Apart from age, sex, ATA risk score, and stage, we did not correct our results for other secondary variables known to be associated with the development or behavior of PTC such as clinical data (e.g., comorbidities, previous ionizing radiation exposure, and medications) or histopathological data (e.g., tumor size, PTC subtypes, histological aggressiveness, focality, and invasiveness). In addition, the statistical power to detect miRNA expression differences across less common PTC subtypes—such as oncocytic, columnar cell variant, etc.—was limited and could have been improved with a larger cohort. However, this was mainly due to the relatively high prevalence of the conventional subtype and a relatively low occurrence of other histological variants, which is consistent with other population-level observations [68,69]. It is also worth mentioning that our sample size is still larger than that used in many similar studies in this regard. Plus, this study focuses on individual miRNA expression variations and does not investigate inter-miRNA interactions or the combined effects of the miRNAs on PTC development. In addition, our data would have been even more reliable if we had been able to carry out repeated measurements of those miRNAs found to be significant in this study. However, this was not feasible due to the sample size limitations mentioned above. Also, we used FFPE tissue samples instead of fresh tissues or FNAB samples, which may impact the real clinical value of our results. Furthermore, we recognize that our study lacks functional validation experiments which would be crucial for gaining a deeper understanding of the relation between PTC pathogenesis and the identified differentially expressed miRNAs. Therefore, conducting functional validation tests (in vivo and/or in vitro) of the relevant miRNAs, such as knockdown animal experiments, would be highly beneficial. Lastly, an extension of our investigation in the future to multiple centers would definitely help verify our conclusions by avoiding potential patient selection biases from the same geographical region.

## Figures and Tables

**Figure 1 ijms-25-09362-f001:**
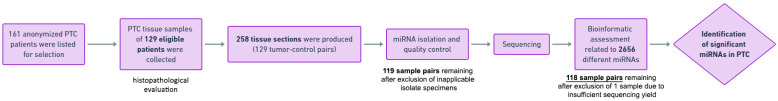
This workflow diagram illustrates the steps of the miRNA analysis of PTC patients. We reviewed 161 anonymized PTC cases of the tissue archives, from which 129 were selected as eligible based on histopathological evaluation. A total of 258 thyroid tissue samples (129-129 tumor and control samples, respectively) related to these cases were then collected and subjected to sectioning. Then, the sections underwent miRNA isolation and quality control of RNA concentrations, leading to the exclusion of samples being evaluated as inapplicable isolate specimens. The remaining samples were then subjected to sequencing, after which a bioinformatic and statistical assessment was conducted on the data in the context of 2656 different miRNA types in total. Bioinformatic evaluation led to the further exclusion of 1 sample pair (both tumor and control) due to insufficient sequencing yield detected. Finally, we were able to establish those miRNAs which show significantly different expression patterns in PTC and non-PTC tissues related to 118 patients in total.

**Figure 2 ijms-25-09362-f002:**
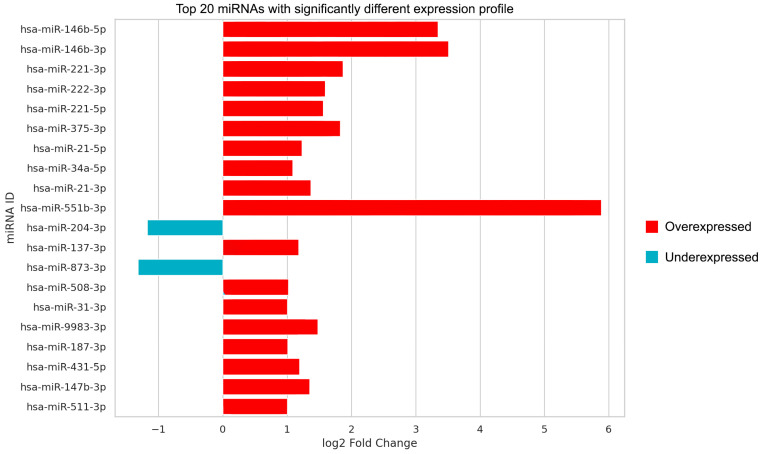
The bars of this chart present the log_2_ fold change of the top 20 miRNAs (named on the vertical axis) selected based on their significantly different expression profiles between the cancer and control groups. Bars that extend to the right of the zero line (red) show overexpression of the particular miRNA in tumor tissue, while those to the left (blue) indicate underexpression.

**Figure 3 ijms-25-09362-f003:**
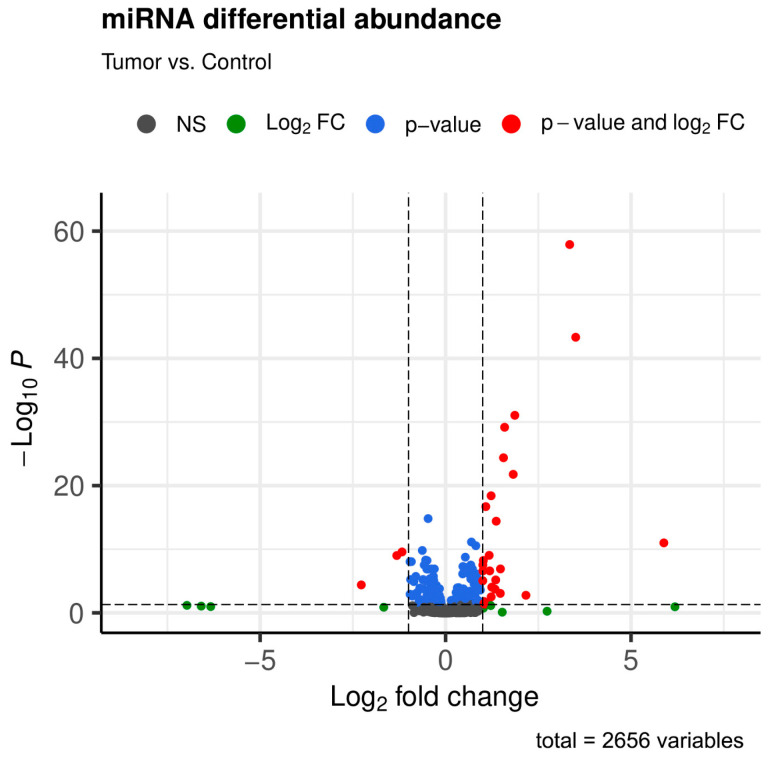
This volcano plot illustrates the different expressions of the miRNAs. On the horizontal axis, the log_2_ fold change is represented, highlighting the magnitude of expression deviations. The vertical axis illustrates the negative logarithm of the *p*-value (−log_10_P), reflecting the statistical significance of the expression change related to each miRNA. Dots above the horizontal threshold line (blue and red) represent miRNAs that pass the significance criterion. Dots to the right or left of the vertical threshold lines (red) indicate not only high significance levels but also a substantial overexpression or underexpression of the corresponding miRNAs, respectively. Dots below the horizontal threshold line represent miRNAs with large fold changes that are not statistically significant (green) or miRNAs that do not meet any of the threshold values (gray).

**Figure 4 ijms-25-09362-f004:**
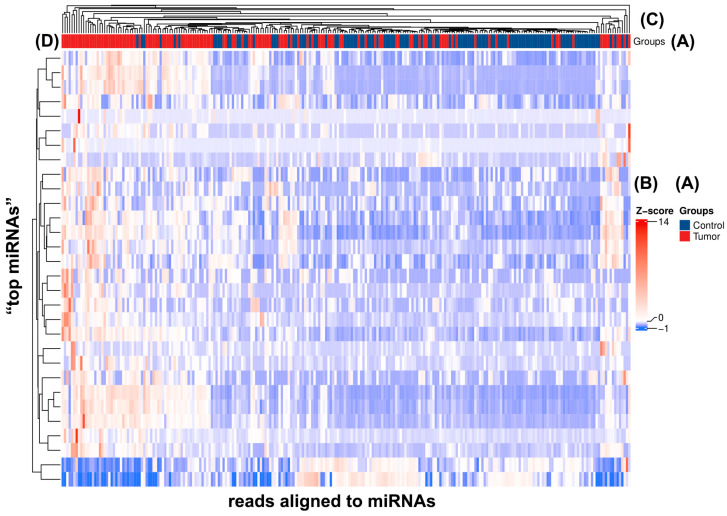
In this heatmap, the rows correspond to the “top miRNAs” (n = 30) of this study, selected based on their significantly different expression levels between tumor (red) and control (blue) groups categorized by histopathological characteristics. (**A**) Each column represents one tissue sample (n = 236) subjected to molecular analysis. The color intensity within each cell reflects the Z-score derived from the normalized number of reads aligned to significant “top miRNAs”, with more red shades indicating higher expression and more blue indicating a lower expression pattern of the particular miRNA of the row. (**B**) Hierarchical clustering is applied to both “top miRNAs” and samples of the two groups, as shown by the black branches, grouping similar expression profiles together. The vertical dendrogram (black lines on the vertical axis) illustrates the hierarchical clustering of “top miRNAs”, categorizing them based on the similarity in their expression patterns across all samples, while the horizontal dendrogram (black branches on the horizontal axis) represents the hierarchical clustering of samples, highlighting that the samples with similar miRNA expression profiles tend to fall into the same (either control or tumor) group (**C**,**D**).

**Figure 5 ijms-25-09362-f005:**
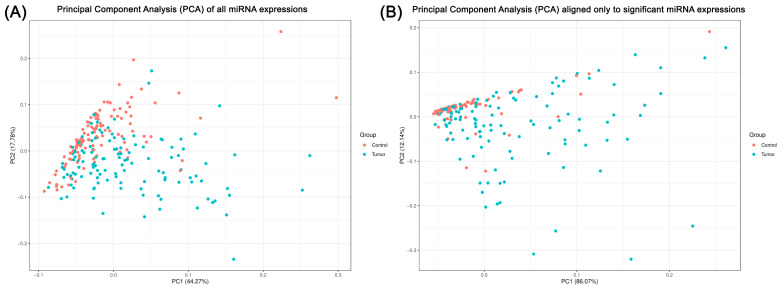
Comparative principal component analysis of miRNA expressions in tumor and control samples. In plot (**A**), a PCA of all miRNA expressions tested is shown, with the horizontal axis representing Principal Component 1 (PC1), which accounts for 44.27% of the variance, and the vertical axis representing Principal Component 2 (PC2), accounting for 17.78% of the variance. Variables of the control group are marked in red and the tumor group in blue, indicating moderate separation along PC1, suggesting differential expression patterns between the two states. Plot (**B**) however displays a PCA focused exclusively on miRNA expressions found to be significant previously, with PC1 explaining a dominant 86.07% of the variance and PC2 accounting for 12.14%. Here, the separation between the two groups is more pronounced along PC1, indicating an explicit distinction in the expression profiles. The juxtaposition of these two plots highlights that specific miRNAs (marked as significant) contribute mostly to the molecular variance between the tumor and non-tumor conditions. The comparison illustrates the utility of focusing on significant miRNAs for a more targeted understanding of the molecular background of PTC.

**Figure 6 ijms-25-09362-f006:**
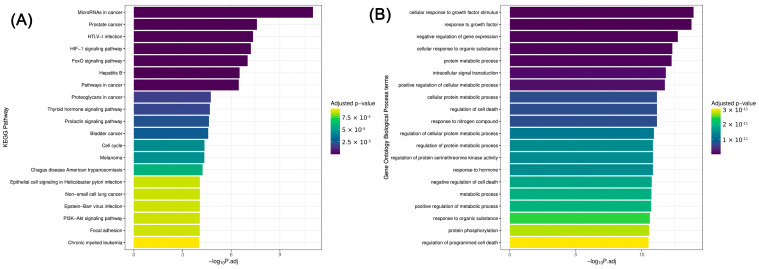
KEGG and Gene Ontology (GO) enrichment analyses (ORA—over-representation analysis) based on statistically significant (*p* ≤ 0.05) miRNAs of this study. Associations were found between the miRNA expression patterns in PTC marked as significant and the molecular patterns of pathways (**A**) listed in the KEGG database as well as biological processes (**B**), cellular components (**C**), and molecular functions (**D**) listed in the GO database. Based on the strength of significance, the plot visualizes the top 20 molecular patterns of the KEGG and GO databases showing potential correlation with PTC. Each bar represents a pathway, a biological process, a cellular component, or a molecular function of these databases (vertical axes), with the length of the bar reflecting the significance of a possible association with PTC as indicated by the −log10 of the adjusted *p*-value (P.adj) (horizontal axes). The color gradient conveys the adjusted *p*-value, transitioning from yellow (less significant) to dark purple (more significant). The data suggest that these molecular patterns (**A**–**D**) may be influenced by the same miRNAs as the development and/or progression of PTC.

**Figure 7 ijms-25-09362-f007:**
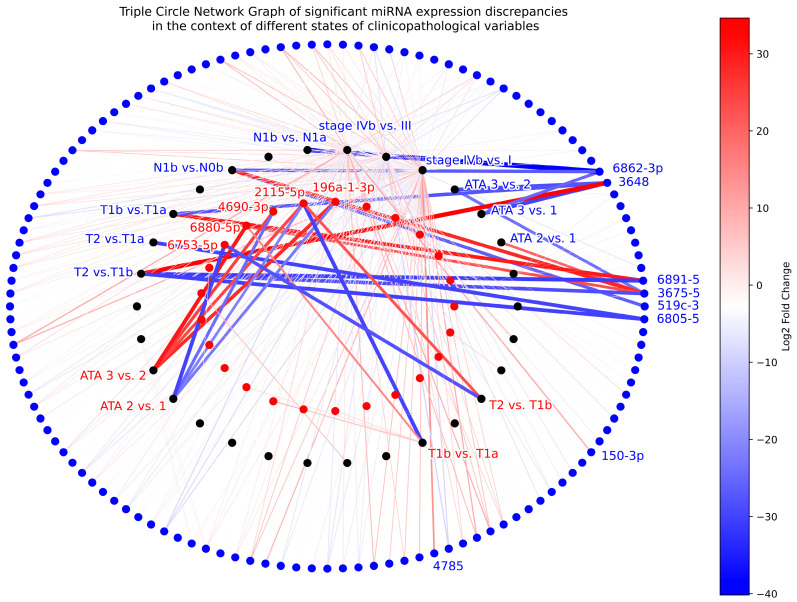
Triple circle network graph illustrating the most relevant miRNA expression differences between certain states of the examined clinicopathological variables such as age, sex, ATA risk, and stages (TNM and AJCC eighth edition) (middle circle, black nodes). Differentially expressed miRNAs within the control samples are represented as blue nodes (outermost circle), whereas they are indicated as red nodes in the context of tumor samples (innermost circle). All red and blue nodes represent a significant change (*p* < 0.05) in miRNA expression in relation to at least one clinicopathological variable. The significant associations are indicated by lines, with blue indicating negative changes and red indicating positive changes in miRNA expressions. The color gradient of the lines from blue to red represents the log_2_ fold change (log_2_FC) of miRNA expressions, with darker shades representing greater expression differences and thus stronger links. To provide a clear and uncluttered visual representation of the network structure, the graph is devoid of any node labels related to associations with log_2_FC values between 10 and −10.

**Table 1 ijms-25-09362-t001:** PTC subtypes in the studied cohort. In our cohort of 118 patients with papillary thyroid carcinoma (PTC), the two most frequent subtypes are the conventional subtype, accounting for 81.36% of cases, and the follicular subtype, comprising 13.56% of cases. These findings highlight the dominance of the conventional subtype in PTC incidence and the significant presence of the follicular subtype, underscoring the variability within PTC presentations.

PTC Subtype	n=	%
Conventional	96	81.35
Follicular subtype	16	13.56
Oncocytic	4	3.39
Columnar cell	1	0.85
Warthin-like	1	0.85
All PTC subtypes	118	100

**Table 2 ijms-25-09362-t002:** The magnitude and significance of the expression deviations between tumor and non-tumor tissue samples corresponding to the “top 30 miRNAs”. The log_2_ fold change (FC) indicates the average expression level changes in the listed miRNAs. The standard error (SE) reflects the variability of the log_2_ FC estimates. Statistical significance was assessed using false discovery rate (FDR)-corrected *p*-values, with significance set at *p* < 0.05.

miRNA	log_2_ FC	SE	FDR-Corrected *p*
hsa-miR-21-5p	1.227	0.130	3.969 × 10^−19^
hsa-miR-21-3p	1.364	0.163	3.926 × 10^−15^
hsa-miR-31-3p	1.004	0.164	3.018 × 10^−8^
hsa-miR-34a-5p	1.084	0.120	1.999 × 10^−17^
hsa-miR-187-3p	1.005	0.176	2.520 × 10^−7^
hsa-miR-221-5p	1.560	0.144	4.208 × 10^−25^
hsa-miR-221-3p	1.866	0.153	8.969 × 10^−32^
hsa-miR-222-5p	1.035	0.353	0.01644
hsa-miR-222-3p	1.591	0.135	6.766 × 10^−30^
hsa-miR-137-3p	1.175	0.175	9.120 × 10^−10^
hsa-miR-375-3p	1.823	0.178	1.694 × 10^−22^
hsa-miR-376a-5p	1.475	0.381	8.764 × 10^−4^
hsa-miR-431-5p	1.189	0.209	2.520 × 10^−7^
hsa-miR-511-3p	1.003	0.201	8.883 × 10^−6^
hsa-miR-146b-5p	3.345	0.202	1.294 × 10^−58^
hsa-miR-146b-3p	3.507	0.245	4.798 × 10^−44^
hsa-miR-508-3p	1.017	0.159	5.819 × 10^−9^
hsa-miR-510-5p	1.147	0.415	0.0251
hsa-miR-514a-5p	1.229	0.350	0.0031
hsa-miR-556-5p	1.333	0.312	1.913 × 10^−4^
hsa-miR-551b-5p	2.166	0.589	0.0017
hsa-miR-551b-3p	5.884	0.797	1.006 × 10^−11^
hsa-miR-147b-3p	1.351	0.267	6.713 × 10^−6^
hsa-miR-1277-5p	1.064	0.415	0.0405
hsa-miR-514b-5p	1.245	0.278	9.230 × 10^−5^
hsa-miR-4695-3p	1.034	0.389	0.0317
hsa-miR-9983-3p	1.479	0.253	1.247 × 10^−7^
hsa-miR-204-3p	−1.175	0.170	2.675 × 10^−10^
hsa-miR-206	−2.273	0.488	4.060 × 10^−5^
hsa-miR-873-3p	−1.316	0.197	9.781 × 10^−10^

## Data Availability

Some datasets generated and analyzed during the current study are not publicly available but are available from the corresponding author [Ármós R.] on reasonable request.

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
