# Peer review of "MicroRNA Profiling in Papillary Thyroid Cancer"

_ijms, 2024, doi:10.3390/ijms25179362_

Round 1

Reviewer 1 Report

Comments and Suggestions for Authors

The study examined miRNA expression patterns in papillary thyroid carcinoma (PTC) tissue samples, identifying significant differences between tumor and control tissues. Of 2656 miRNAs analyzed, 30 showed notable dysregulation. The research highlights miRNAs with diagnostic potential for PTC and provides insights into associated biological pathways and clinicopathological variables.

My Comments

1.      How were the 129 papillary thyroid carcinoma (PTC) patients selected from the initial cohort of 161 anonymized cases? Were there specific inclusion or exclusion criteria beyond histopathological evaluation?

2.      Out of 129 initial patients, the final cohort includes 118 PTC patients. Does this sample size provide adequate power for statistical analyses, especially for detecting differences across various PTC subtypes?

3.      The distribution of PTC histological subtypes is skewed, with a strong dominance of the conventional subtype (81.35%) and a small representation of other subtypes (e.g., follicular, oncocytic). How might this uneven distribution affect the generalizability of the findings, particularly for underrepresented subtypes?

4.      How were the non-tumor control samples collected and verified to be free of any pathological changes? Were they matched to tumor samples in terms of patient demographics, location within the thyroid, or other relevant factors?

5.      The study mentions that 582 miRNA types were not expressed at all in thyroid tissue. How was this threshold for "no expression" determined, and could there be any significance to these non-expressed miRNAs in thyroid cancer?

Author Response

Thank you for your review. Please see our responses in the attachment.

Reviewer 2 Report

Comments and Suggestions for Authors

Overall, this manuscript represents a thorough analysis of miRNA expression in papillary thyroid carcinoma (PTC), based on a large sample size including a wide range of miRNAs. The manuscript combines bioinformatics analysis with clinicopathological factors, thus providing insights into the potential clinical implications of their dysregulation in PTC and therefore they could represent diagnostic and prognostic biomarkers for such a neoplasm. Interestingly, the authors identify 30 miRNAs whose expression is significantly different in PTC compared to healthy thyroid tissue, including novel associations between some miRNAs and PTC. Also, their findings rely on a rigorous statistical analysis. However, the authors do not provide a functional validation of differentially expressed miRNAs and therefore the precise mechanisms by which these miRNAs contribute to PTC pathogenesis must be elucidated. Moreover, the authors only focus on individual miRNA expression changes but do not extensively discuss their potential interactions and their combined effects and should therefore mention this limitation of their work. Moreover, the authors should specify the retrospective nature of their study and should perform a spell and punctuation check and moderate English editing. Lastly, I would like the authors to discuss future directions, for example the relationship between miRNAs expression and tumor microenvironment (doi: 10.1038/s41598-023-42941-1), immune cell infiltration (doi: 10.3389/fonc.2021.755097; doi: 10.3389/fgene.2021.710412; https://doi.org/10.1002/cncy.22224) and the role of miRNAs as serum/plasmatic biomarkers of PTC (doi: 10.3389/fmed.2023.1139362) with possible parallelisms with other neoplasias (https://doi.org/10.1186/s12885-018-4442-2).

Comments on the Quality of English Language

Moderate English editing is needed

Author Response

(The authors gave the same response as above.)
